# Second Version of a Mini-Survey to Evaluate Food Intake Quality (Mini-ECCA v.2): Reproducibility and Ability to Identify Dietary Patterns in University Students

**DOI:** 10.3390/nu12030809

**Published:** 2020-03-19

**Authors:** María Fernanda Bernal-Orozco, Patricia Belen Salmeron-Curiel, Ruth Jackelyne Prado-Arriaga, Jaime Fernando Orozco-Gutiérrez, Nayeli Badillo-Camacho, Fabiola Márquez-Sandoval, Martha Betzaida Altamirano-Martínez, Montserrat González-Gómez, Porfirio Gutiérrez-González, Barbara Vizmanos, Gabriela Macedo-Ojeda

**Affiliations:** 1Bachelor of Nutrition, Centro Universitario de Ciencias de la Salud (CUCS), Universidad de Guadalajara (UdeG), Sierra Mojada 950, Building “N”, Colonia Independencia, Guadalajara, Jalisco, ZC 44340, Mexico; fera_18@yahoo.com.mx (M.F.B.-O.); salmeronbelen@gmail.com (P.B.S.-C.); jacky.pa.51@gmail.com (R.J.P.-A.); jfog97@live.com.mx (J.F.O.-G.); nuta.nayeli@yahoo.com.mx (N.B.-C.); yolanda.marquez@academicos.udg.mx (F.M.-S.); martha.altamirano@academicos.udg.mx (M.B.A.-M.); mgonzalez_investmed@hotmail.com (M.G.-G.); bvizmanos@yahoo.com.mx (B.V.); 2Doctorate in Traslational Nutritional Sciences, CUCS, UdeG, Juan Díaz Covarrubias and Salvador Quevedo y Zubieta, Building “C”, Colonia Independencia, Guadalajara, Jalisco, ZC 44340, Mexico; 3Department of Human Reproduction, Growth and Child Development, CUCS, UdeG, Hospital 320, Colonia El Retiro, Guadalajara, Jalisco, ZC 44100, Mexico; 4Translational Nutrigenetics and Nutrigenomics Institute, CUCS, UdeG, Sierra Mojada 950, Building Q, first floor, Colonia Independencia, Guadalajara, Jalisco, ZC 44340, Mexico; 5Mathematics Department, Centro Universitario de Ciencias Exactas e Ingenierías, UdeG, Blvd. Marcelino García Barragán 1421, Guadalajara, Jalisco, ZC 44430, Mexico; 6Department of Public Health, CUCS, UdeG, Sierra Mojada 950, Building “N”, Colonia Independencia, Guadalajara, Jalisco, ZC 44340, Mexico

**Keywords:** diet patterns, food intake quality, reproducibility, eating behavior, food assessment, diet, undergraduate health students

## Abstract

Evaluation of food intake quality using validated tools makes it possible to give individuals or populations recommendations for improving their diet. This study’s objective was to evaluate the reproducibility and ability to identify dietary patterns of the second version of the Mini Food Intake Quality Survey (Mini-ECCA v.2). The survey was administered using a remote voting system on two occasions with four-week intervals between administrations to 276 health science students (average age = 20.1 ± 3.1 years; 68% women). We then performed a per-question weighted kappa calculation, a cluster analysis, an ANOVA test by questionnaire item and between identified clusters, and a discriminant analysis. Moderate to excellent agreement was observed (weighted κ = 0.422–0.662). The cluster analysis identified three groups, and the discriminant analysis obtained three classification functions (85.9% of cases were correctly classified): group 1 (19.9%) was characterized by higher intake of water, vegetables, fruit, fats, oilseeds/avocado, meat and legumes (healthy food intake); group 2 (47.1%) frequently consumed both fish and unhealthy fats (habits in need of improvement); group 3 (33%) frequently consumed sweetened beverages, foods not prepared at home, processed foods, refined cereals and alcohol (unhealthy food intake). In conclusion, the Mini-ECCA v.2 has moderate to excellent agreement, and it is able to identify dietary patterns in university students.

## 1. Introduction

Diet plays a determinant role in either the maintenance or deterioration of health. More specifically, the intake of certain foods, food groups or nutrients has been proven to have an influential effect on the prevention, development and treatment of a number of chronic non-communicable diseases [1,2]. In light of this reality, dietary quality evaluations have become fundamentally important in today’s clinical and epidemiological research [3].

The term dietary quality is often used to identify the components of a healthy, balanced and nutritious diet and adapt them to the needs of specific populations with the aim of optimizing health [4]. One way of determining dietary quality is through the use of indices that make it possible to categorize an individual’s diet after comparing that person’s actual and recommended intake levels [5,6]. Some of the most internationally recognized indices are the Healthy Eating Index (HEI) [7,8,9,10,11,12], the Alternative Healthy Eating Index (AHEI), the Diet Quality Index (DQI) [13,14] (this one has been adapted in different countries and age groups), the Diet Diversity Score (DDS) [15] and the Healthy Diet Indicator (HDI) [16], among others. In Mexico, the Mexican Diet Quality Index (ICDMx) has gained significant recognition [17]. 

The administration, analysis and data interpretation of these tools are time-consuming, as they entail the use of mechanisms such as food surveys, 24-h reminders, food records or semiquantitative food frequency questionnaires in order to study dietary factors such as adequacy, balance, variety, safety, and so on [5,18]. With 24-h recalls and semiquantitative food frequency questionnaires, cognitive functions such as conceptualization (the ability to make a mental construction of a particular amount of food not present) and memory (the ability to remember the amount ingested) may lead to confusion in the subject and contribute to potential sources of error in estimating quantities, which do not allow the accuracy of the tool used to be assessed with a risk of under-reporting bias to be present [19]. 

The need of surpassing these application difficulties inherent in the evaluation of the diet led us to seek to build an easy and quick tool to apply with visual support, to be able to make both group and individual diagnosis of eating habits and decisions of intervention for the improvement of quality consumption.

Taking this into consideration, we identified other indices that have been developed which evaluate only food intake quality, as opposed to diet quality. In particular, they analyze the extent to which adherence to food intake recommendations, the intake of specific food groups or a mixture of both has either a positive or negative effect on health [20]. These tools have few question items, are precise and easy to answer and require short administration and interpretation times [21]. Examples of such indices are those that measure adherence to the Mediterranean diet [22,23,24,25].

In Mexico, a tool called the Mini Food Intake Quality Survey (Mini-ECCA), whose structure was similar to that of indices which measured adherence to the Mediterranean diet and included photographs as a visual reference to estimate quantity, as they present lower food quantity estimation error than other visual support tools [19], provided a rapid assessment of food intake quality as indicated by adherence to intake recommendations. This survey produced good levels of reproducibility (ρ = 0.713, *p* < 0.001; ICC = 0.844, 95% CI, 0.793–0.883) and moderate levels of agreement in terms of food intake quality classification (κ = 0.545, 95% CI, 0.484 to 0.606, *p* < 0.001), and thus proved to be suitable for dietary evaluation and guidance purposes [26]. This is, as far as we know, the only short survey assessing food quality or adherence to national (Mexican) [17,27,28] and international guidelines (at that time) [1,29,30,31,32,33,34,35,36] in Latin America.

However, several areas of opportunity were identified to improve the tool. Specifically, due to the low number of items, a one-point variation between test and retest scores could have an impact on the food intake quality classification, so it was suggested changing the Mini-ECCA’s response option format from a dichotomous one to the type found on a Likert scale, and changing the way this survey is scored. Furthermore, our tool did not include alcohol intake, a topic that also must be evaluated in food intake quality. Another limitation of the first version of the tool was that some questions needed to be reworded so people unfamiliar with nutritional terminology could easily understand them like fats and sweetened beverages, and no other analyses besides reproducibility were considered [26]. In addition, given the benefits associated with reviewing, updating, improving and validating existing tools in ways that will give health professionals and authorities more accurate data for decision-making [4], a second version of the Mini-ECCA was developed. 

On the other hand, in this second version of the Mini-ECCA, the reproducibility analysis is complemented by the evaluation of the tool’s ability to identify dietary patterns. This perspective provides additional elements to examine the overall diet and not the consumption of specific nutrients or food groups, as it is suggested that dietary patterns could have a more real relationship with positive or adverse health effects [37]. The objective of this study was to assess the levels of reproducibility of the Mini-ECCA v.2 and its ability to identify dietary patterns in university students. The reason for choosing university students is because identifying the quality of their food consumption must allow us to contribute to favor an environment of healthy options, so that they could improve their dietary intake and, therefore, their health.

## 2. Materials and Methods 

### 2.1. Modification of the Mini-ECCA Questionnaire

A group of experts who participated in the instrument’s original design, all of whom are health professionals with clinical and nutritional research experience (M.F.B-O.; F.M.-S.; M.B.A.-M.; M.G.-G.; B.V.; G.M.-O.), with a group of young nutritionists (R.J.P.-A.; N.B.-C. and J.F.O.-G.) analyzed, discussed and approved changes to the tool based on the areas of opportunity above mentioned.

As a result of this process (Table 1), changes were made to recommended intake amounts of some foods and food groups, such as fish and sweetened beverages. Several questions were reformulated to make them clearer and more specific (like in sweetened beverages, fish, sweets and desserts, fats and legumes), and with more emphasis on food intake frequency (daily or per week); Questions that were not in the original instrument were also added, specifically on the intake of avocados, oilseeds and alcoholic beverages. In addition, a Likert scale replaced the dichotomous (yes/no) response options found in the previous version. This change was designed to make the options more indicative actual food intake behaviors, which tend not to be strictly dichotomous in nature.

The Mini-ECCA v.2 thus consists of 14 questions focused on assessing food and beverage (alcoholic and non-alcoholic) intake. The questions are based on a review of relevant literature on the subject, including Mexican [38,39] and international nutritional recommendations [1,35,36,40,41,42,43,44,45,46,47,48,49,50,51,52,53,54,55,56,57] (Table 1). For each question, three or four answer choices are given, generally on a Likert scale (see Appendix A: Mini-ECCA v.2 survey). The questionnaire uses photographs as a visual reference to estimate quantity; thus, images are shown simultaneously with questions during the administration of the Mini-ECCA v.2 (see Presentation S1: Mini-ECCA’s visual aid for food quantity estimation). 

### 2.2. Study Population and Design

A study was carried out between October 2017 and May 2018 to determine the reproducibility and the ability to identify dietary patterns of the Mini-ECCA v.2 in university students enrolled in the Nutrition (NUT), Medicine (MED) and Dentistry (OD) programs at the University of Guadalajara’s Health Sciences Center. Subjects were students of both sexes of varying ages who were enrolled in different university grade levels. In the case of NUT students, we randomly selected a group from each semester, from first to sixth, and invited them to participate (October-December 2017). In the case of MED and OD students, we invited to participate all groups attending the subject called “Healthy lifestyles promotion” (March-May 2018). Those groups whose teachers allowed us to invite students were selected. Finally, we included volunteers who agreed to participate and gave their informed consent. Excluded were pregnant or lactating women, as were individuals diagnosed with any illness requiring a diet different from that of healthy people, due to the significant differences in nutritional needs associated with such conditions. In addition, subjects on vegetarian diets were excluded since the instrument considers a healthy omnivorous diet as its benchmark. Additionally, participants who did not take part in the questionnaire’s second administration were excluded.

A minimum sample of 140 participants was determined based on criteria in scientific literature suggesting the inclusion of 10 subjects for each question on the instrument, which will be analyzed in reproducibility studies [66].

The study adhered to Mexican and international ethical principles for research involving human subjects which are designed to safeguard the dignity, autonomy, rights, privacy and confidentiality of research participants [67]. Additionally, the study’s protocol was approved by the Research and Research Ethics Committees of the University’s Health Sciences Center (registration number: CI-01519).

### 2.3. Data collection Procedures and Strategies

After obtaining the required university authorizations, the administration schedule for the Mini-ECCA v.2 was planned based on the times when the selected group’s teacher and fieldwork researchers would be available. The procedure followed for each administration is described below.

Administration 1: The researchers arrived to the selected group’s classroom on the day and at the time agreed with their teacher. The objective, implications and importance of participation in the study were explained to the volunteer subjects verbally, and a document with detailed information on the project was provided. After explanations, volunteers signed an informed consent form and filled out a self-reported medical history in which they gave contact, socioeconomic, anthropometric (weight and height that allowed us to calculate Body Mass Index classified according to WHO criteria), and pathological (personal and family) data. They also completed the International Physical Activity Questionnaire (Short Form IPAQ-SF) [68] as part of the self-reported medical history procedure. The Mini-ECCA v.2 was then administered using the SunVote^®^ system (software v.3.1.0.20, M52 Plus response clicker). Researchers trained in the use of this remote voting system explained the procedure for using it to participants. The students answered each of the questions in the Mini-ECCA v.2 using remote control units following the projection of questions, answer options and visual support for estimating food quantities (see Presentation S1: Mini-ECCA’s visual aid for food quantity estimation). First administration’s duration was 30 min in total (15 min only for Mini-ECCA, the other 15 min were used for answering the other personal information).

Administration 2: One month after the first administration, the second administration of the questionnaire was carried out on the same group of students. Second administration’s duration was 15 min and was performed the same as the first administration. Finally, personal results (from both the first and second administrations) together with a thank you message were e-mailed to each participant a few days following the second administration.

### 2.4. Statistical Analysis

The data obtained were statistically analyzed using SPSS^®^ software (version 25.0, SPSS Inc., Chicago, IL). Qualitative variables are expressed as frequency (percentage) and quantitative variables as averages (standard deviations). 

The reproducibility of the Mini-ECCA was determined using the agreement between the answers of each question in the Mini-ECCA v.2 based on Cohen’s weighted kappa—which ranges from 1 (perfect disagreement) to 1 (perfect agreement)—and categorized according to Landis and Koch criteria [69]. McNemar test was used to determine whether changes in categorical responses between test and retest were significant.

The determination of the instrument’s ability to identify dietary patterns was carried out by the following analyses. First, a cluster analysis was performed using Ward’s method for non-standardized variables. Cluster analysis is a multivariate method that, in this case, aggregates individuals in relatively homogeneous subgroups according to the frequency of the food groups consumed [37]. Subsequently, an analysis of simple variance (ANOVA) was performed for each item in the questionnaire to compare the groups identified in the cluster analysis. Finally, a discriminant analysis was performed. These analyses aimed to verify that the groups identified in the cluster analysis had significant differences in food intake quality or dietary patterns. A value of *p* < 0.05 was considered significant.

## 3. Results

### 3.1. Description of the Population

Mini-ECCA v.2 was initially administered to 397 students (all the students agreed to participate). From this group, the following were excluded: 8 who reported following a vegetarian diet; 3 who reported being pregnant or lactating; 4 who were psychology students (for not being a representative sample of their major); and 7 who did not complete the test. Of the 375 remaining students, 96 did not take the retest and another 3 did not complete the second test (26.4% attrition rate). Thus, this study analyzed a final total of 276 participants who completed both survey administrations.

The average age of the population was 20.1 ± 3.1 years (*n* = 270, 6 did not respond). Of the total number of participants, 68% were women and 56% were medical students. In addition, 80% did not work and almost all of them (98%) had no financial dependents. Approximately 37% reported moderate physical activity, and their average number of hours spent sitting per day was 8.3 ± 3.9 (*n* = 223). According to self-reported data on disease history, 30.4% of participants reported being overweight/obese, while 2.5% reported having some form of heart disease, 2.2% reported having asthma and 1.1% reported having other diseases such as hypothyroidism or polycystic ovary syndrome (data not shown). 

Other self-reported data included an average weight of 64.4 ± 13.7 kg (*n* = 268), an average height of 166.7 ± 9.3 cm (*n* = 267) and an average BMI of 23.1 ± 3.8 kg/cm^2^ (*n* = 267). The majority of the sample (65.2%) had normal weight. For more details on the characteristics of the population, see Table 2. 

### 3.2. Reproducibility and Concordance of the Mini-ECCA v.2

The analysis of all of the Mini-ECCA v.2’s questions between the test and retest showed moderate concordance (weighted κ = 0.422–0.585) in questions about the intake of water, vegetables, fish, fruits, oilseeds and/or avocado, foods not prepared at home, industrialized snacks, sweets and/or desserts, legumes, sweetened beverages and type of cereal most frequently consumed. Three questions showed excellent agreement (weighted κ = 0.606–0.662): intake of alcoholic beverages, type of fat used most often, and type of meat. It should be noted that a significant change between test and retest was detected in two questions: water intake (*p* = 0.041) and intake of sweets or desserts (*p* = 0.035). For details, see Table 3.

### 3.3. Ability to Identify Dietary Patterns of the Mini-ECCA v.2: Cluster and Discriminant Analysis 

In the cluster analysis, three groups were identified: the first group included close to a fifth of the participants (*n* = 55, 19.9%) who consumed a diet that was, based on a discriminant analysis, subsequently classified as healthy; the habits of the second group (*n* = 130, 47.1%), which accounted for almost half of the sample, were considered as in need of improvement (intake of some healthy and other unhealthy foods); the third group (*n* = 91, 33%) comprised participants with unhealthy food choices.

After the three groups were identified, an ANOVA was performed for each item in the Mini-ECCA v.2 questionnaire in order to verify that this grouping would be capable of identifying differences in food intake quality. This analysis of variance produced a value of *p* < 0.05 in all cases (see Document S1: Differences in the response option average for each question in the Mini-ECCA v.2 by cluster).

Table 4 shows three classification functions (one per group) obtained in the discriminant analysis, as well as the items included in each group. The highest value for each variable was the one considered for assignment to the group (shown in boldface). As shown in the table, group 1 was characterized by a higher intake of water, vegetables, fruits, fats, oilseeds/water, meats and legumes (healthy food intake). Group 2 was characterized by a healthier intake of fish combined with a higher intake of unhealthy fats (habits in need of improvement). Group 3 was characterized by a higher intake of sweetened beverages, foods not prepared at home, processed foods, refined cereals and alcoholic beverages (unhealthy food intake). 

By means of the discriminant analysis, two discriminant functions with a value of *p* < 0.05 were also obtained. Table 4 shows only the values obtained from the first discriminant function as well as its coefficients, as it represents a canonical correlation of 0.84957, with a relative percentage of 89.45%. The highest values of the discriminant function indicate the characteristics that make the groups different. In this regard, the variable that best explains the differences between groups is vegetable intake (coefficient = 0.50719), followed by water intake (coefficient = 0.403538). The variables that explain the differences between groups to a lesser extent are fresh or frozen fish intake (coefficient = 0.0655855) and processed food intake (coefficient = −0.125812).

The discriminant analysis determined that the groups found were adequately classified through application of the classification functions (Table 5), as 85.9% of the total cases were correctly grouped. In addition, a correct classification has been identified in 96.4% of cases in the healthy food intake group (Group 1), 78.5% in the habits in need of improvement group (Group 2) and 90.1% in the unhealthy food intake group (Group 3). 

## 4. Discussion

The Mini-ECCA v.2 survey obtained moderate to excellent agreement [69] between the test and the retest for each of its items. It also was able to identify dietary patterns, with 85.9% of the total cases having been correctly classified in the three groups identified in the cluster analysis. This new version of the questionnaire includes items covering the intake of alcohol and the most consumed types of fat. It also provides more answer options for each question, which range from daily intake (always) to zero intake (never), in addition to intermediate options such as almost always or sometimes, which may reflect intake better than items with dichotomous answers [70].

The Mini-ECCA v.2 is similar in some ways to other surveys that evaluate dietary adherence (especially to the Mediterranean diet) [21,22,29,71]: number of items, administration’s duration, selected food groups or foods inclusion, answer options, and so on. However, it is important to notice that these surveys were designed to examine dietary intake and association with some health outcomes like cardiometabolic risk [21,22,29,71], which is not the aim of Mini-ECCA v.2. Our tool places special emphasis on evaluating the quality of foods consumed, either healthy or non-healthy [72].

The Mini-ECCA v.2 used the cluster analysis and the discriminant analysis method. This procedure complements the reproducibility analysis and helps determine if the questionnaire can identify dietary patterns and if they present differences between them. This method is practical and does not require parameters other than the variables found in the survey such as biochemical markers or other dietary surveys (although these associations are also important for validation purposes and could be performed in subsequent studies.) These analyses have been used in studies from other health areas, like in psychology [73,74], and since food intake is an eating behavior, these analyses are suitable for our proposed survey. Thus, this is one of the main contributions of this study. 

A discriminant analysis yields the coefficients needed to generate classification functions, which when calculated allow the quality of the food intake of an individual or a population to be determined based on answers from the Mini-ECCA v.2. In order to obtain the classification of an individual’s food intake quality for any one of the three groups identified in this study, the three classification functions were applied to each subject, hence multiplying the answer options of each question of the Mini-ECCA v.2 by the constants of the corresponding column (Table 4). After obtaining the results of the three equations, the quality of the subject’s dietary intake was classified for the group for which the highest figure was obtained (Document S2: Equations for the interpretation of Mini-ECCA v.2’s results). Thus the interpretation of the first version of the Mini-ECCA went from having four possible results (very low, low, good and excellent) to having three (healthy food intake, habits in need of improvement, unhealthy food intake). It is important to note that subjects who fell into the “unhealthy food intake” classification may consume some foods mentioned in the survey with a frequency that would score as healthy, but as these behaviors occur less frequently than unhealthy ones, they were classified under “unhealthy food intake.”

In other studies, dietary pattern analysis has led also to the identification of both healthy and non-healthy food intake. This identification may lead to specific actions of food intake improvement in populations where these dietary patterns have been identified [75,76,77]. In the case of university health students, dietary patterns of healthcare students and professionals was described and associated with sociodemographic, lifestyle, anthropometric and biochemical characteristics in a previous study [75]. In this study, dietary patterns were identified through a principal component analysis of 25 food groups of a semiquantitative food frequency questionnaire. These dietary patterns were “Traditional Westernized”, “Healthy” and “Animal protein and alcoholic beverages”. As evidenced in that study, healthy and non-healthy dietary patterns can be identified in this particular population.

When analyzing the answers to each of the questions in the Mini-ECCA v.2, agreement is observed as generally being moderate to excellent (weighted κ = 0.422–0.662). However, it should be noted that in the case of two questions (water intake and intake of sweets and desserts), intake quality showed statistically significant improvement in the retest. This may be because, although there was a one-month interval between administrations of the Mini-ECCA v.2, participants may have become more aware of their food/beverage intake behavior, meaning they could have either modified their behavior or given answers that they believed to be more acceptable [78]. In light of this possibility, it is very important to emphasize to the survey’s subjects that their honesty is needed in order to determine what the quality of their food intake really is and offer them appropriate guidance. It should be noted that the degree of agreement on the first version of the Mini-ECCA was moderate (weighted κ = 0.484–0.606). It should also be remembered that the first version consisted of dichotomous questions, and the time interval between test–retest administrations was shorter (15 days). This shows that despite the change of options and the increase in the time between one administration and the next, the Mini-ECCA v.2 produced a higher level of agreement. 

With respect to food and drink intake habits, it is noteworthy that only 52.2% of the test sample reported drinking at least 1.5 liters of water per day, while 45.7% of the sample reported consuming it 1 to 3 times per week including one or more glasses of sweetened beverages. According to data from the 2006 National Health and Nutrition Survey (ENSANUT), the average water intake of the general population was less than 900 mL [79], a quantity similar to that reported in a study of liquid intake in Latin America published in 2018, where the average daily intake ranged from 150 to 900 mL per day [80]. Regarding sweetened beverage intake, it is well known that Mexico’s levels are higher than those of any other country. This is particularly true in the case of soft drinks, for which average intake is just over 500 mL per day [80]. Despite the fact that current recommendations advise against consuming these drinks [35,38], their intake continues to be very common, even among health science students. It should be noted that since most recent ENSANUT reports do not provide an analysis of water intake in milliliters (it only asks whether or not water is consumed), subsequent surveys will have to go back to include more precise beverage intake measurements.

Regarding the intake of vegetables, the most frequently given answers were sometimes (35.9%) and almost always (40.2%); in the case of fruit intake, the results were similar with almost always (33.3%) and always (38.4%) being the most answered options. Only 20%–40% of the surveyed students comply with the recommendations for daily intake of fruits and vegetables. These results were similar to those reported by Muñoz-Cano et al. [81], where fewer than 20% of respondents consumed vegetables and fruits more than three times per week [81].

Regarding fresh or frozen fish intake, the most frequently given answer was sometimes (44.2%), indicating that this population does not generally consume food from this group. This conclusion was confirmed by the question on the type of meat consumed most often, where fish was answered by less than 5% of the respondents. In contrast, when asked to indicate the type of meat they habitually consume, 53.6% of respondents answered chicken, while 42.0% answered red meat. The above results coincide with the annual intake reported by the OECD for Mexico, where chicken was the most consumed type of meat (30 kg/per person), followed by pork (14.7 kg/per person) and beef (8.9 kg/per person) [82]. The average fish intake per person in Mexico is estimated at 10.5 g per day [83], and Latin America considered to be the region of the world with the lowest intake of this food category [84].

In addition, 52.9% of those surveyed indicated sometimes as their frequency of consuming industrialized products, while 20.7% said they almost always consume them. Similarly, 52.9% of participants reported sometimes consuming other products such as snacks or instant foods. Similar results were observed in the consumption of desserts and sweets, as 43.1% indicated that they sometimes consume them. Results from ENSANUT 2018 found that 35.4% of the population over 20 years of age eats snacks, sweets and desserts on a daily basis [85].

All these previous studies indicate that the quality of food intake among health science students is not very different than that of the rest of the population. This finding underscores the urgency of making future health professionals aware of the need for them to have healthier dietary habits, since they will be tasked with the future prevention and treatment of diseases in the population, and congruence is an essential part of positively influencing patient behavior. To achieve this aim, it will also be important to promote healthy habits in university campus environments by offering “healthy menus” in student dining facilities, discouraging the on-campus sale of sweetened beverages, and so forth.

Limitations of this study include a significant number of drop-outs in the administration of the retest (25.8%), as is common in follow-up studies. However, these losses did not affect the minimum number of subjects expected for validation. Another limiting factor is that the Mini-ECCA is not suitable for administration to vegetarian or vegan subjects. The design and validation of tools for these populations, as well as for age groups such as children and adolescents, therefore represents an opportunity for the future. Yet another limitation is that the use of health science students as volunteers may lead to the conclusion that this group is significantly different from the rest of the population, or that their knowledge of health sciences may bias their responses. However, and as shown in the analysis of each question in the Mini-ECCA v.2, food intake quality in this study is very similar to that found in previously published data on the general population [85].

As strengths, we can highlight the method used for dietary pattern identification, which is easier and practical than the analysis of other dietary surveys and a more practical assessment of adherence to a healthy or to a non-healthy food intake. In addition, it allows an easier follow-up of patients and of our academic population in order to help them to better realize food and beverages consumption for a healthier lifestyle. The ability of identifying these three patterns, will allow us to characterize our university population and promote changes to a better pattern consumption.

## 5. Conclusions

The Mini-ECCA v.2 is a tool which can be administered quickly and easily and produces moderate to excellent levels of concordance and able to identify dietary patterns. It is useful for evaluating the quality of food intake in university students and classifying it into healthy food intake, habits in need of improvement and unhealthy food intake categories. This survey should also be effective for studying the general adult population, as no major differences were identified with respect to intake levels reported in health and nutrition surveys conducted at the national level. It may also make health science students more aware of the effect that their own dietary habits may have on their ability to be positive role models and thus better prepare them to care for the health of the general population. In this regard, making university campus environments more conducive to healthy behaviors will also be important.

## Figures and Tables

**Table 1 nutrients-12-00809-t001:** Recommendations on which the changes to each question in the Mini-ECCA v.2 were based.

Food/Food Group	Evidence	Mini-ECCA v.1 Question	Mini-ECCA v.2 Question	Changes Made from the First Version
Water	The intake of at least 6 glasses (1500 mL) per day is recommended [38] to meet daily fluid needs and achieve an optimal state of hydration to help maintain metabolism and normal physiological functions such as thermoregulation, excretion, transport, etc. [39].	Do you drink at least 1.5 liters of water per day?	Do you drink at least 1.5 liters of water every day (Monday to Sunday)?	The wording was changed to emphasize that intake days should be from Monday to Sunday.
Fruits and vegetables	It is recommended to consume at least five whole pieces or portions equivalent to 400 g per day [1], because of this group’s significant fiber, micronutrient and phytochemical content which contributes to health maintenance [51,52]. However, since the recommendation is for two food groups considered collectively, the total amount was divided evenly (200 g for fruits and 200 g for vegetables).	Do you consume at least 200 g of fruit per day? Do you consume at least 200 g of cooked or raw vegetables per day?	Do you consume at least 200 g of fruit every day (Monday to Sunday)? Do you consume at least 200 g of vegetables every day (Monday to Sunday)?	The wording was changed to emphasize that intake days should be from Monday to Sunday.
Fish	It is recommended to consume fish at least twice per week (or two portions per week) because of its long-chain polyunsaturated fatty acid content, which is associated with reduced cardiovascular risk [53,54]. With a serving defined as 100 g (3.5 ounces), the recommendation calls for an intake of 200 g per week [55].	Do you eat fresh or frozen fish (100 g) at least one day per week?	Do you consume at least 200 g of fresh or frozen (not canned) fish per week?	The previous intake amount was 100 g.
Sweetened beverages	The intake of sweetened beverages should be occasional and in small portions, as they are associated with increased energy intake, higher body weight, increased risk of type 2 diabetes mellitus, and because they have little or no nutritional value [38]. It was calculated that the maximum tolerable limit would be one glass (250 mL) or less per day, an amount which would not exceed the recommended maximum of 10% of total calories from added sugars [35].	Do you consume four or more sweetened beverages per week?	How many times per week do you consume one or more cans (or glasses) of sweetened beverages?	In the previous version, an intake of 4 or more drinks per week was mentioned with no reference to intake volume.
Oils and fats	A total fat intake comprising 15%–35% of daily energy is recommended (less than 7% from saturated fatty acids, up to 10% from polyunsaturated fatty acids and up to 20% from monounsaturated fatty acids) [56] in order to deliver a supply of essential fatty acids and fat-soluble vitamins that meets the needs of most individuals, as well as to ensure optimal health and prevent the development of cardiovascular diseases [57,40].	What type of fat do you most frequently consume during the week?	What oil or fat-based ingredient do you use most often on a weekly basis to prepare your meals?	This question was divided into two categories: oils/fats and oilseeds/avocado.
Oilseeds and avocado	It is recommended to consume 20–30 g of nuts, seeds and olives [41] due to the decrease in cardiovascular risk factors associated with their daily intake, and particularly in light of the antioxidant properties of this food group which may inhibit or delay atherogenic processes [42,43,58]. No precise avocado intake recommendations have been established, but a minimum intake of half of an avocado per day has been found to be effective in reducing cardiovascular risk [44,45].	---	Do you consume at least 30 g of oilseeds or one-half of an avocado every day (Monday to Sunday)?	This question did not appear in the previous version. Oilseeds had been included in the question on oils and fats.
Meat	Red meat intake should be limited to less than twice per week because of its association with an increased incidence of cardiovascular disease and cancer [46]. The intake of skinless poultry and fish is recommended as an alternative [41,47].	What type of meat do you consume most often?	What type of meat do you consume most often on a weekly basis?	In the new version, intake was specified as weekly in order to make the question more precise.
Processed foods	Processed foods are characterized by their high sodium content. It is recommended for adults to maintain daily sodium intake below 2 g (5 g of salt) [36]. In addition, the intake of processed foods should be limited because of their link to cancer [46,48].	Do you eat processed foods two or more days per week?	Do you consume processed foods (fried foods, sausages, packaged meals ready to heat and serve) 2 or more times per week?	The wording was changed. In the previous version, intake was measured in days per week.
Meals consumed away from home	Eating out more frequently has been associated with increased weight gain due to the higher energy content of the food consumed and the larger portions served [49]. It is recommended to limit meals consumed away from home to less than three times per week [50].	Do you eat foods not prepared at home three or more days per week?	Do you consume food not prepared at home 3 or more times per week?	The wording was changed. In the previous version, intake was measured in days per week.
Sweets and desserts	It is recommended for free sugars not to exceed 10% of total daily energy intake, although it has been suggested that additional benefits may result from reducing total energy intake from sugars to 5% or less [35]. Consequently, the intake of sweets and desserts should be limited due to their high sugar content.	Do you consume sweets or commercial desserts two or more days per week?	Do you eat dessert foods (cookies, creme caramel (flan), rice pudding, cakes) or sweets (hard candy, popsicles, chocolates) 2 or more times per week?	Only commercially produced desserts or sweets were mentioned in the previous version. Examples of desserts and sweets were added.
Legumes	It is recommended to consume two to four portions of legumes per week, as they are an affordable source of fiber, protein and other nutrients associated with the prevention of various diseases [59,60]. Considering that a portion is equivalent to 150–200 g of cooked legumes, recommendations range from 300–800 g per week [41]. Due to variability in quantity, a minimum recommended quantity was established (300 g).	Do you eat legumes at least three days per week (300 g per week)?	Do you consume at least 300 g of legumes per week?	An intake frequency of 3 times per week was used in the previous version.
Cereals	It is recommended for more than 50% of daily cereal intake to be whole grain. Because of its high content of fiber and other nutrients, this food has been shown to protect against various chronic non-communicable diseases, contribute to gastrointestinal health maintenance and help control body weight [34,61]. It is therefore advisable to consume three or more servings of whole grains per day and to reduce the intake of refined and processed cereals [34,62].	What cereals do you consume most often?	What kind of cereals do you consume most often on a weekly basis?	Intake frequency was stated as weekly.
Alcoholic beverages	While avoiding alcohol intake has been recommended [63,64], a recent review of 83 prospective studies concluded that 100 g of alcohol per week is the intake threshold associated with the lowest risk of mortality [63]. Based on national and international guidelines, a limit of one drink per day was established for women and two for men [34,38,65].	---	If you are a man, do you consume more than 2 alcoholic beverages per day? If you are a woman, do you consume more than 1 alcoholic beverage per day?	This question did not appear in the previous version.

**Table 2 nutrients-12-00809-t002:** General characteristics of the population (*n* = 276).

Variable	Frequency	Percentage
Sex		
Male	88	31.9
Female	188	68.1
Career		
Nutrition	86	31.2
Medicine	169	61.2
Odontology	21	7.6
Worked		
No	221	80.1
Yes	51	18.5
No answer given	4	1.4
Had financial dependents ^1^		
No	266	96.4
Yes	6	2.2
No answer given	4	1.4
Had financial support ^2^		
No	149	54.8
Yes	118	43.4
No answer given	9	1.8
Physical Activity Level		
Low	96	34.8
Medium	101	36.6
High	79	28.6
BMI Classification ^3^		
Underweight	20	7.5
Normal	174	65.2
Overweight	59	22.1
Obese	14	5.2

^1^ Refers to people who are financially dependent on the student (e.g., children, parents, etc.); ^2^ Refers to whether the student has any scholarship or other financial assistance (such as that provided by parents) to help pay for their studies; ^3^
*n* = 267.

**Table 3 nutrients-12-00809-t003:** Concordance in the questions in the Mini-ECCA v.2.

Question.	Answer Options	Test	Retest	Total Agreement	Weighted Kappa (95% CI)	McNemar Test (*p* Value)
		*n* (%) ^1^	*n* (%) ^1^	*n* (%) ^2^		
1. Do you drink at least 1.5 liters of water per day?	A. Never	8 (2.9)	1 (0.4)	1 (12.5)	0.585 (0.572–0.598)	0.041
	B. Sometimes	64 (23.2)	55 (19.9)	37 (57.8)		
	C. Almost always	67 (24.3)	76 (27.5)	32 (47.8)		
	D. Always	137 (49.6)	144 (52.2)	115 (83.9)		
2. Do you consume at least 200 g of cooked or raw vegetables per day?	A. Never	16 (5.8)	9 (3.3)	4 (25.0)	0.478 (0.465–0.490)	0.220
	B. Sometimes	99 (35.9)	94 (34.1)	60 (60.6)		
	C. Almost always	111 (40.2)	112 (40.6)	67 (60.4)		
	D. Always	50 (18.1)	61 (22.1)	30 (60.0)		
3. Do you eat fresh or frozen fish (100 g) at least one day per week?	A. Never	73 (26.4)	71 (25.7)	52 (71.2)	0.576 (0.562–0.589)	0.877
	B. Sometimes	122 (44.2)	130 (47.1)	87 (71.3)		
	C. Almost always	45 (16.3)	43 (15.6)	21 (46.7)		
	D. Always	36 (13.0)	32 (11.6)	19 (52.8)		
4. Do you consume one or more glasses (can) of sweetened beverages per week?	A. Never	37 (13.4)	46 (16.7)	27 (73.0)	0.576 (0.562–0.589)	0.054
	B. 1 to 3 times	126 (45.7)	139 (50.4)	94 (74.6)		
	C. 4 to 6 times	59 (21.4)	44 (15.9)	20 (33.9)		
	D. Daily	54 (19.6)	47 (17.0)	26 (48.1)		
5. Do you consume at least 200 g of fruit per day?	A. Never	7 (2.5)	6 (2.2)	1 (14.3)	0.516 (0.504–0.529)	0.525
	B. Sometimes	71 (25.7)	70 (25.4)	44 (62.0)		
	C. Almost always	92 (33.3)	103 (37.3)	53 (57.6)		
	D. Always	106 (38.4)	97 (35.1)	70 (66.0)		
6. What type of fat do you consume most frequently on a weekly basis?	A. Monounsaturated	60 (21.7)	59 (21.4)	46 (76.7)	0.662 (0.654–0.671)	0.800
	B. Polyunsaturated	201 (72.8)	201 (72.8)	182 (90.5)		
	C. Saturated	7 (2.5)	10 (3.6)	4 (57.1)		
	D. Do not know	8 (2.9)	6 (2.2)	4 (50.0)		
7. Do you consume at least 30 g of oilseeds or 1/2 of an avocado per day?	A. Never	29 (10.5)	27 (9.8)	13 (44.8)	0.483 (0.470–0.496)	0.270
	B. Sometimes	127 (46.0)	141 (51.1)	94 (74.0)		
	C. Almost always	77 (27.9)	76 (27.5)	37 (48.1)		
	D. Always	43 (15.6)	32 (11.6)	17 (39.5)		
8. Do you eat foods not prepared at home three or more days per week?	A. Never	55 (19.9)	37 (13.4)	28 (50.9)	0.579 (0.566–0.592)	0.061
	B. Sometimes	123 (44.6)	139 (50.4)	89 (72.4)		
	C. Almost always	55 (19.9)	59 (21.4)	24 (43.6)		
	D. Always	43 (15.6)	41 (14.9)	30 (69.8)		
9. What type of meat do you consume most often?	A. Red meat	116 (42.0)	109 (39.5)	87 (75.0)	0.606 (0.594–0.618)	0.539
	B. Chicken	148 (53.6)	154 (55.8)	123 (83.1)		
	C. Fish	12 (4.3)	13 (4.7)	7 (58.3)		
10. Do you eat processed foods two or more days per week?	A. Never	46 (16.7)	44 (15.9)	23 (50.0)	0.458 (0.445–0.470)	0.823
	B. Sometimes	146 (52.9)	154 (55.8)	105 (71.9)		
	C. Almost always	57 (20.7)	52 (18.8)	22 (38.6)		
	D. Always	27 (9.8)	26 (9.4)	14 (51.9)		
11. Do you consume sweets or commercially produced desserts two or more days per week?	A. Never	22 (8.0)	33 (12)	12 (54.5)	0.422 (0.409–0.436)	0.035
	B. Sometimes	119 (43.1)	131 (47.5)	78 (65.5)		
	C. Almost always	71 (25.7)	64 (23.2)	23 (32.4)		
	D. Always	64 (23.2)	48 (17.4)	26 (40.6)		
12. Do you eat legumes at least three days per week (300 g per week)?	A. Never	18 (6.5)	17 (6.2)	10 (55.6)	0.495 (0.482–0.509)	0.946
	B. Sometimes	56 (20.3)	55 (19.9)	25 (44.6)		
	C. Almost always	74 (26.8)	77 (27.9)	31 (41.9)		
	D. Always	128 (46.4)	127 (46.0)	92 (71.9)		
13. What type of cereals do you consume most often?	A. Whole grain	118 (42.8)	135 (48.9)	88 (74.6)	0.427 (0.414–0.441)	0.200
	B. Minimally processed	120 (43.5)	111 (40.2)	77 (64.2)		
	C. Processed and ultra-processed	38 (13.8)	30 (10.9)	16 (42.1)		
14. If you are a man, do you consume more than 2 alcoholic beverages per day? If you are a woman, do you consume more than 1 alcoholic beverage per day?	A. Never	169 (61.2)	165 (59.8)	148 (87.6)	0.636 (0.625–0.647)	0.492
	B. Sometimes	79 (28.6)	83 (30.1)	52 (65.8)		
	C. Almost always	20 (7.2)	14 (5.1)	5 (25.0)		
	D. Always	8 (2.9)	14 (5.1)	4 (50.0)		

^1^ Data are presented as frequency (percentage of the total); ^2^ It refers to frequency (percentage of test subjects who showed agreement on the retest or answered the same in both administrations).

**Table 4 nutrients-12-00809-t004:** Coefficients of the classification and discriminant functions for food intake quality by group.

Variable	Classification Function Coefficients by Group			Discriminant Function Coefficients
	1 Healthy Food Intake	2 Habits in Need of Improvement	3 Unhealthy Food Intake	Function 1
Water intake	**6.2692**	5.59019	4.18735	0.403538
Vegetable intake	**8.63136**	7.59581	5.37525	0.50719
Fresh/frozen fish intake	1.67472	**1.98176**	1.50061	0.0655855
Sweet drink intake	2.24428	2.92271	**3.13341**	−0.15818
Fruit intake	**7.30975**	6.57655	4.95411	0.39329
Most consumed fats or oils	4.86969	**6.71207**	6.62932	−0.172202
Oilseeds or avocado intake	**4.63935**	3.18834	2.54328	0.333013
Intake of foods not prepared at home	1.30334	1.67774	**2.47401**	−0.23877
Most consumed type of meat	**7.18377**	5.77848	5.34274	0.198137
Intake of processed foods	1.73401	2.00778	**2.42516**	−0.125812
Sweets or commercially produced desserts	1.09702	2.01592	2.19824	−0.186589
Intake of legumes	**4.31863**	4.31195	3.02602	0.304913
Most consumed type of cereals	2.51518	3.05353	**3.88732**	−0.210954
Alcoholic beverages	−0.460071	−0.245971	**0.486413**	−0.17358
CONSTANT	−74.8948	−69.9333	−55.1233	
Eigenvalue				2.59421
Canonical correlation				0.84957 *
Wilks’ Lambda				0.213051
Chi-squared				412.0688 **
DF				28

The coefficients that were considered for the classification of each group are shown in boldface, with the highest value of each variable being the one considered for the group assignment. DF: degrees of freedom. ** *p* < 0.05. * *p* value < 0.001.

**Table 5 nutrients-12-00809-t005:** Correct classification of cases according to the identified clusters.

Classification Group	*n* (%) ^1^	*n* (%) Correctly Classified Cases ^2^
Group 1: Healthy food intake	55 (19.9)	53 (96.4)
Group 2: Habits in need of improvement	130 (47.1)	102 (78.5)
Group 3: Unhealthy food intake	91 (33.0)	82 (90.1)
Total	276 (100.0)	237 (85.9)

^1^ Data are presented as frequency (percentage of the total); ^2^ It refers to frequency (percentage of subjects from the middle column).

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
