# Peer review of "Second Version of a Mini-Survey to Evaluate Food Intake Quality (Mini-ECCA v.2): Reproducibility and Ability to Identify Dietary Patterns in University Students"

_nutrients, 2020, doi:10.3390/nu12030809_

Round 1

Reviewer 1 Report

Line 55 – Suggest revising reference 7 to a newer manuscript discussing the HEI. Suggest Krebs-Smith, Susan M., et al. "Update of the healthy eating index: HEI-2015." Journal of the Academy of Nutrition and Dietetics 118.9 (2018): 1591-1602.

Suggest reviewing all dietary indices and using the most up to date versions of each survey

Line 81 – Can the authors also briefly describe the “areas for opportunity”

Line 81/Table 1 – How likely is it that the general public knows if they’re using “mostly polyunsaturated fats” for example? Do the authors have any evidence to suggest that the public actually know what this is?

Line 81/Table 1 – Why were grams of fruits and vegetables used rather than servings or pieces? My understanding is that grams isn’t used in the federal guidelines.

Table 1 – Can the authors add in a column with the Mini-ECCA v.1 items to make it easier to compare

Line 90 – Please provide references to Mexican and International guidelines that were used as a basis to create the Mini-ECCA

After reading the introduction its unclear why a second version of the Mini-ECCA was developed? The first Mini-ECCA had good levels of reproducibility and levels of agreement. The authors should provide further justification for why.  

The introduction also reads very similar to the original Mini-ECCA manuscript’s introduction. Suggest revising to avoid duplication Bernal-Orozco, M.; Badillo-Camacho, N.; Macedo-Ojeda, G.; González-Gómez, M.; Orozco-Gutiérrez, J.; Prado-Arriaga, R.; Márquez-Sandoval, F.; Altamirano-Martínez, M.; Vizmanos, B. Design and Reproducibility of a Mini-Survey to Evaluate the Quality of Food Intake (Mini-ECCA) in a Mexican Population. Nutrients 2018, 10, 524.

Line 132 – How long did administration take?

Line 134 – Was administration conducted the same as in Time 1?

In the discussion the authors compare the Mini-ECCA v.2 to other surveys which evaluate dietary patterns such as the Mediterranean Diet Score, the Mediterranean Diet Adherence Score, and the Cardioprotective Mediterranean Diet Questionnaire. This surveys were designed to examine dietary intake and association to some health outcome like cardiometabolic risk. My understanding of the Mini-ECCA v.2 is that it was not developed in the same manner. The foods chosen for inclusion on the Mini-ECCA were not designed with some health protective effect. Can the authors justify why they provide a lengthy description of the other surveys and how they relevantly compare to the Mini-ECCA?

I’m also confused why the authors don’t compare their dietary measure to other surveys that measure dietary intake in Latin America as I think this would be more relevant.

The supplemental materials referenced in lines 95, 97, and 132 were not available to be reviewed

Reviewer 2 Report

In general, the manuscript is well-written and authors have analyzed their data using standard statistical techniques. My main concern relates to an over-interpretation of the study’s findings. Two are the components generally considered in the assessment of a questionnaire’s performance: (a) its reproducibility, often reflecting random errors and (b) its relative validity, which encompasses comparisons with another method (possibly “the gold standard”) and provides input for sources of systematic errors. In this analysis, the authors assessed the reproducibility of the questionnaire (test-retest), but their protocol did not purposefully include an assessment of the relative validity.
It should be noted, however, used the cluster analysis and discriminant analysis method, could have been considered as an assessment that goes beyond reproducibility.
Having that said, I would propose to the authors to downgrade the interpretation of their findings and solely indicate that their results point to the reproducibility of the questionnaire tested. Statements such as “ ...its validity levels are acceptable for the evaluation of food intake quality in university students” are not well substantiated.

Keywords: I suggest to add more keywords

Introduction: The introduction was informative and provided a clear overview. Whilst I appreciate that this paper is investigating the test-retest reliability, it is also important to consider the validity of such retrospective subjective tools. It may be important to outline the under-reporting bias associated with such methods here to contextualise the research.

Methods: Participants - students were not randomly selected. Authors need to provide more the selection criteria. Could the choice of these participants have had any impact on observations?
Did the Authors also use other statistics to assess reproducibility and validity, e.g. Bland-Altman index to assess the reproducibility (test vs. retest)?

Results: The formatting of your tables may need reworking as it was difficult to follow and some data were misaligned to the associated variable. Check it, please.

Discussion: Some of the contents in this section should be addressed in the introduction order to provide enough background information on retrospective tools. Note: Contents of introduction and discussions are interchangeable.

For better readability, I suggest to include Appendix in supplementary.

Round 2

Reviewer 1 Report

Thank you to the authors for their thorough revision of the manuscript. The changes made by the authors have addressed all of my concerns. I have no further comments. 

Reviewer 2 Report

This manuscript has been significantly improved in this form.
A good job was done.
I suggest adding a questionnaire "Mini Encuesta de Calidad del Consumo Alimentario Mini-ECCA" in the Supplementary Materials in Spanish and English. This will allow their use by other researchers.

I can now support its application.

Best regards.